cognition/neuroscience

body perception, spatial cognition, multisensory integration

**Authors for correspondence:**
Arvid Guterstam
e-mail: arvid.guterstam@gmail.com
H. Henrik Ehrsson
e-mail: henrik.ehrsson@ki.se

# Duplication of the bodily self: a perceptual illusion of dual full-body ownership and dual self-location

Arvid Guterstam[1,2], Dennis E. O. Larsson[2], Joanna Szczotka[2] and H. Henrik Ehrsson[2]

[1]Department of Psychology, Princeton University, Princeton, NJ, USA
[2]Department of Neuroscience, Karolinska Institutet, Stockholm, Sweden

 AG, 0000-0002-3694-1318

Previous research has shown that it is possible to use multisensory stimulation to induce the perceptual illusion of owning supernumerary limbs, such as two right arms. However, it remains unclear whether the coherent feeling of owning a full-body may be duplicated in the same manner and whether such a dual full-body illusion could be used to split the unitary sense of self-location into two. Here, we examined whether healthy human participants can experience simultaneous ownership of two full-bodies, located either close in parallel or in two separate spatial locations. A previously described full-body illusion, based on visuo-tactile stimulation of an artificial body viewed from the first-person perspective (1PP) via head-mounted displays, was adapted to a dual-body setting and quantified in five experiments using questionnaires, a behavioural self-location task and threat-evoked skin conductance responses. The results of experiments 1–3 showed that synchronous visuo-tactile stimulation of two bodies viewed from the 1PP lying in parallel next to each other induced a significant illusion of dual full-body ownership. In experiment 4, we failed to find support for our working hypothesis that splitting the visual scene into two, so that each of the two illusory bodies was placed in distinct spatial environments, would lead to dual self-location. In a final exploratory experiment (no. 5), we found preliminary support for an illusion of dual self-location and dual body ownership by using dynamic changes between the 1PPs of two artificial bodies and/or a common third-person perspective in the ceiling of the testing room. These findings suggest that healthy people, under certain conditions of multisensory perceptual ambiguity, may experience dual body ownership and dual self-location. These findings suggest that the coherent sense of the bodily self located at a single place in space is the result of an active and dynamic perceptual integration process.

# 1. Introduction

Having two arms and two legs and being physically located at one single spatial location are fundamental subjective experiences that most of us take for granted. However, certain neurological conditions can lead to profound disturbances in our senses of limb ownership and self-location. Brain lesions or focal epileptic activity sometimes result in the experience of having an extra arm or leg, so-called supernumerary phantom limbs [1–4], or even full-blown out-of-body experiences involving the feeling of dual self-location [5–9], demonstrating that our natural senses of body ownership and self-location depend on certain intact neural circuits. Experimental work in healthy human participants has shown that the perceptual illusion of owning an artificial hand [10], or supernumerary artificial hands [11–14], can be achieved by means of multisensory stimulation. Although limb ownership illusions can be extended to involve an artificial full-body viewed from the natural (first-person) perspective with the use of head-mounted displays (HMDs) and correlated multisensory stimulation [15–18], it remains unclear whether the corresponding full-body illusion can be elicited for supernumerary bodies. The aim of this study was to examine whether the full-body illusion can be elicited for two bodies at the same time and whether such a supernumerary dual full-body illusion can be used to induce the illusion of dual self-location, similar to the patient experiences described in the neurological literature.

Two previous studies have reported attempts to experimentally induce dual body ownership and/or self-location [19,20]. In [19], 13 participants wore HMDs connected to a camera attached to a humanoid robot and used a joystick to guide the robot's movements along a predetermined path that ended up with the participants viewing themselves from the perspective of the robot. The results of a questionnaire indicated that participants experienced being located in both the robot and their real body, and slightly more so than when they had control over the robot's movements versus a control condition in which they had not. However, the questionnaire did not assess the degree of ownership of the seen real body and contained no statements to control for suggestibility or task compliance, making the results difficult to interpret. Furthermore, the study did not quantify ownership and self-location with any established behavioural (e.g. proprioceptive drift [21]) or physiological measurement (e.g. threat-evoked skin conductance responses (SCRs) [22]). Although this preliminary observation indicates that the healthy human brain has the capacity to represent dual self-location, it is unclear whether a dual full-body ownership illusion was induced and whether this visuo-movement approach can be used to manipulate ownership and self-location in a predictable manner.

Heydrich *et al*. [20] adapted the full-body illusion set-up reported in [23], where a virtual body is viewed from a distance from a third-person perspective (3PP), to a dual-body setting. Participants wore HMDs and observed from a distance of a couple of metres an object repeatedly touching the backs of either two virtual copies of themselves (experiment 1) or virtual avatars (experiment 2) [20]. Although synchronous visuo-tactile stimulation resulted in a higher degree of self-identification with the virtual bodies, participants did not report any robust feeling of having supernumerary full-bodies. The median rating on a 7-point Likert scale ranging from 1 (I don't agree at all) to 7 (I totally agree) of the key statement 'It seemed as if I might have more than one body' was 3 in experiment 1 and approximately 1.5 in experiment 2 for both the synchronous and asynchronous conditions, showing that participants on average disagreed with this statement (had people on average agreed more than they disagreed, one would expect a median score greater than 4). Thus, a robust sense of dual full-body ownership was not induced using this paradigm. Furthermore, the study was not designed to address the question of dual self-location. The authors did include a 'spatial location drift task' [23], where participants were asked to indicate their original starting position by walking back to it after being displaced approximately 1.5 m backwards by the experimenter. However, this test was not designed to distinguish between single versus dual self-location, and the results were therefore inconclusive.

The present study had two goals. The first goal was to test the hypothesis that it is possible to elicit an illusion of owning two full-bodies by applying congruent visuo-tactile stimulation of two artificial bodies presented side-by-side and viewed from the first-person perspective (1PP) (experiments 1–3). This hypothesis was based on previous work on full-body illusions using single bodies [15–18] and studies on supernumerary limb illusions [11–14]. The two artificial bodies were placed in close proximity next to each other based on the assumption that this set-up would maximize the probability of eliciting a dual-body ownership illusion and given the importance of multisensory integration within the peripersonal space for own-body perception [24–27]. Illusory dual-body ownership was quantified using questionnaires [10] (experiment 1) and threat-evoked SCRs [13,22,28–32] (experiments 2 and 3).

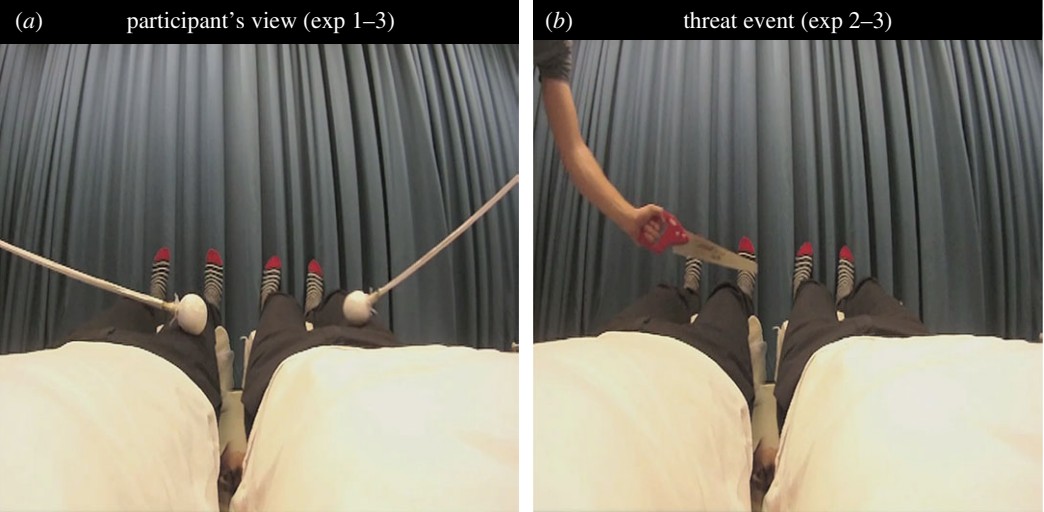

**Figure 1.** Experiments 1–3: Methods. Snapshots from the visual stimuli used in experiments 1–3. The participants lay on a bed with their head tilted forward and wore a set of head-mounted displays where the visual stimuli were presented as highly realistic 3D movies. (*a*) The visuo-tactile stimulation of the two illusory bodies. (*b*) The threat event used in experiments 2 and 3, consisting of a saw sliding over the lower legs of one of the bodies. Only the image of the right eye of the stereoscopic videos is shown.

The second goal was to explore possible ways of creating illusions of dual self-location for illusory bodies placed in different spatial environments, or far apart within a room, by using different relative placements of the two artificial bodies and systematical manipulations of the visual perspective. These manipulations included 'splitting' the visual scene of the 1PP into two distinct spatial environments (experiment 4) and dynamically changing the 1PP and/or 3PP (experiment 5). To quantify self-location in these latter experiments, we used a behavioural self-location task [15,31].

## 2. Material and methods

### 2.1. Participant information

All subjects gave their written informed consent prior to participation, and the Swedish Ethics Review Authority approved all of the experimental procedures. Based on previous studies, we aimed to include 20 participants for the questionnaire and 30 participants for the SCR-based experiments [29]. In experiments with multiple measurements (the questionnaire and self-location task in experiments 4 and 5), we also aimed to include 30 participants. In experiment 1, we included 20 participants (ages 18–27, 13 females, 19 right-handed); in experiment 2, 28 participants (ages 19–31, 19 females, 26 right-handed); in experiment 3, 30 participants (ages 19–33, 16 females, 28 right-handed); in experiment 4, 30 participants (ages 22–55, 17 females, 26 right-handed); and in experiment 5, 30 participants (ages 20–45, 17 females, 28 right-handed) in experiment 5.

### 2.2. Visual stimuli and head-mounted displays

The visual stimuli in all five experiments consisted of pre-recorded three-dimensional (3D) videos of a body (or multiple bodies) being repeatedly touched by an object (see further below and figures 1–3). The videos were recorded at 60 frames per second using two identical cameras (GoPro Hero 5 Black cameras, 1920 × 1080 pixel resolution, GoPro Inc., München, Germany) that were mounted in parallel and capturing the visual 1PP of the body used in the recording (a laboratory member's body in experiments 1–4, and a life-size mannequin in experiment 5), with the cameras positioned at the level of the body's eyes looking down toward the feet. To achieve the desired visual stimuli, such as two identical bodies lying side-by-side (experiments 1–4), separate 3D videos with single bodies were first recorded and then merged using the video editing software Final Cut Pro × (Apple Inc., CA).

The stereoscopic videos were displayed to the participants using HMDs (Oculus Development Kit 2, Facebook Technologies, CA) with a refresh rate of 75 Hz. Participants were lying in a posture identical to the bodies viewed in the video, head slightly tilted with the support of a pillow with the gaze directed

(a) (b)

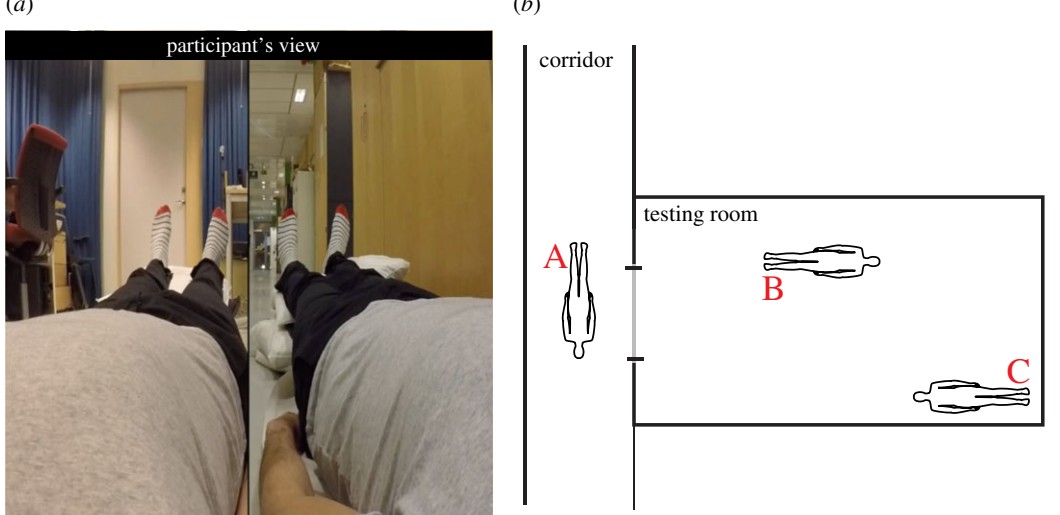

**Figure 2.** Experiment 4: Methods. (a) Snapshot from the visual stimuli in experiment 4. (b) The map used in the behavioural self-location task, in which the letter A indicates the position of the illusory body in the corridor; B, the position of the illusory body in the testing room and C, the position of the participant's real body.

(a) (b)

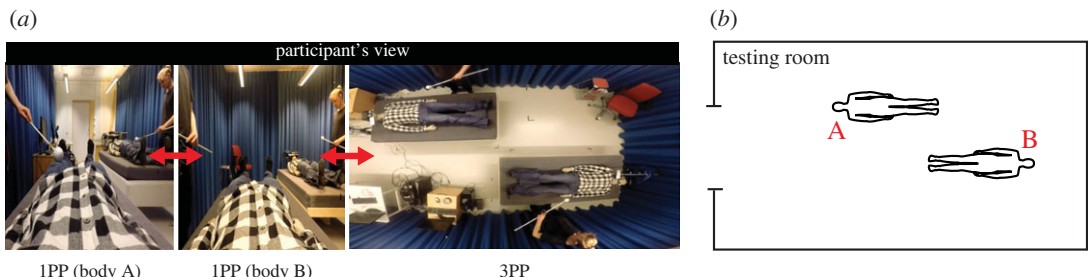

1PP (body A)    1PP (body B)    3PP

**Figure 3.** Experiment 5: Methods. (a) Three snapshots from the visual stimuli in experiment 5, showing the 1PP of body A (also 3PP of body B), the 1PP of body B (also 3PP of body A) and the common 3PP for both bodies A and B. (b) The map used in the behavioural self-location task, in which the letter A indicates the position of body A and the letter B the position of body B.

toward the feet, congruent with the perspective as seen in the video. The participants were instructed to keep their head still throughout the session to avoid any head movements that could potentially reduce the illusion strength. The experimenter wore headphones and listened to a pre-recorded audio track to synchronize the tactile stimulation with the videos. In each video, the bodies (or body) were visibly touched by a white Styrofoam ball (8 cm diameter) attached to the tip of a 1 m long wooden stick. The duration of each touch was 0.7 s, and the time between the offset of one touch and the onset of the next touch was 1.3 s. The touches were applied as continuous strokes along the entire length of the body part on the abdomen, upper legs and the feet in a direction away from the cameras, in the following predetermined random 1 min sequence: A-LL-RL-LL-LF-RL-LL-RF-LF-A-LL-A-LF-RL-LL-RF-LF-A-RF-RL-RF-LL-RL-LF-A-LF-RF-A-RF-RL (A, abdomen; RL, right leg; LL, left leg; RF, right foot; LF, left foot). Only one body part was stimulated at a time. This sequence of touches was identical for all conditions in all experiments and was repeated in the experiments when the duration of one trial was greater than 1 min.

## 2.3. Illusion induction procedure

During testing, the participants wore HMDs and were placed comfortably on a bed in a supine position, with their head tilted forward by approximately 20 degrees supported by pillows, as if looking down at their body. To induce the illusions under investigation, the experimenter used the same object as shown in the videos to deliver touches to the participant's body in synchrony with and on the corresponding body parts as the touches seen in the videos. Because the participants were wearing HMDs, the touches applied to their real body were occluded from view. In the asynchronous

**Table 1.** Questionnaire statements used in experiments 1 and 4.

| |
|---|
| S1: I felt as if I had two bodies (at the same time). |
| S2: I felt the touch of the ball on both bodies at the same time. |
| S3: It felt as if the body to the left were my body. |
| S4: I felt the touch of the ball on the body to the left. |
| S5: It felt as if the body to the right were my body. |
| S6: I felt the touch of the ball on the body to the right. |
| S7: It felt as if I had an invisible body. |
| S8: I could no longer feel my body. |
| S9: I felt the touch of the ball on my back. |
| S10: It seemed as if the touch I were feeling came from somewhere between the body to the left and the body to the right. |

**Table 2.** Questionnaire statements used in experiment 5.

| |
|---|
| S1. It felt as if I were located in two places at the same time. |
| S2. It felt as if I were located in the position of the body facing the door. |
| S3. It felt as if I were located in the position of the body facing the blue curtains. |
| S4. It felt as if I had two bodies (at the same time). |
| S5. It felt as if the body facing the door were my body. |
| S6. It felt as if the body facing the wall with blue curtains were my body. |
| S7. It felt as if I were located in more than two places at the same time. |
| S8. I could no longer feel my body. |

conditions, the touches in the videos were delayed by 1.0 s with respect to the tactile stimulation. Such asynchronous visuo-tactile stimulation is a well-established control condition for full-body illusions [15,18] and allows for comparisons of otherwise equivalent conditions.

## 2.4. Questionnaires

In experiments 1, 4 and 5, we quantified the subjective experiences associated with the illusion using questionnaires that were presented at the end of each repetition (statements adapted from [10,13,29,33,34]). Participants were asked to affirm or deny different statements reflecting potential perceptual effects using a seven-point visual analogue scale that ranged from −3 to +3. The participants were informed that −3 indicated 'I completely disagree', +3 indicated 'I agree completely' and 0 indicated 'I do not know, I can neither agree nor disagree'. The statements were presented one by one on the HMD screen in a random order, and the participants gave a verbal response, which was recorded by the experimenter. The use of this procedure allowed us to conduct an entire experiment session without removing the HMDs from the participants' eyes.

Table 1 summarizes the statements used in experiments 1 and 4, in which statements S1–S2 were designed to capture the feeling of dual body ownership, S3–S4 ownership of the left body, S5–S6 ownership of the right body and S7–S10 served as controls for task compliance. We combined the statements referring explicitly to body ownership and the ones referring to the feeling of touch on the body(-ies) in view (referral of touch), in line with the procedures in [29], because both are key aspects of body ownership illusions [10,18] and body self-perception [35]. Table 2 shows the statements used in experiment 5. Here, S1 was designed to capture the key feeling of dual self-location, S2–S3 the feeling of self-location in the respective illusory bodies, S4 the key feeling of dual body ownership, S5–S6 ownership of the respective illusory bodies and S7–S8 served as controls for task compliance. For our statistical analyses, we used the average ratings of each statement category as the input.

## 2.5. Self-location task

In experiments 4 and 5, we quantified the sense of self-location using a self-location task [15,31] presented on the HMD screen at the end of each repetition. The participants were given a map of the experimental room and were asked to indicate where they had experienced themselves to be located. This map was a proportional representation of the testing room and the corridor (experiment 4), or only the testing room (experiment 5), and had key objects and landmarks indicated (figures 2b and 3b). The participants were asked to rate how strongly they experienced themselves to be located at different spatial locations, indicated by letters, on a continuous visual analogue scale ranging from 0 (I did not experience being located here at all) to 10 (I had a very strong experience of being located here).

## 2.6. Skin conductance responses

In experiments 2 and 3, we measured the threat-evoked SCR, which is an established proxy of illusory body ownership [13,22,28–32]. The SCR was recorded with a Biopac System MP150 and followed standard published guidelines [36]. The two recording electrodes (Biopac TSD203) were attached to the middle phalanges of the index and middle fingers of the participants' right hand, and skin conductance was recorded at a frequency of 100 Hz. The threat stimulus consisted of observing a 50 cm wide saw sliding over the lower legs of one of the mannequins for a duration of 2 s (see figure 1b; electronic supplementary material, Video S1). The threat-evoked SCR was defined as the difference in conductance between the onset time of the threat (i.e. the first moment that the saw entered the participant's visual field) and the peak of the conductance that occurred within 5 s. The SCR data were range-corrected to correct for interindividual variance [36,37]. Each participant's maximum SCR was determined prior to the start of the first experimental block. The participants were instructed to take a deep breath and then hold it for 2 s. Each data point was then expressed as a proportion (ratio) of the range of the SCR response according to the following formula: $SCR_{ratio} = (SCR_{measured\_max} - SCR_{measured\_min})/(SCR_{max} - SCR_{min})$ [37]. If the $SCR_{ratio}$ exceeded 1, a value of 1 was given. The average of all responses for each participant, including those in which no increase in amplitude was apparent, was separately calculated for each condition, and this value was taken as the SCR magnitude. Thereafter, the SCR magnitudes for all of the participants were compared statistically across the different conditions as described in the Results section.

## 2.7. Statistical approach

Because the majority of the datasets did not pass the Kolmogorov–Smirnov test for normality, we decided to use non-parametric tests for all analyses for reasons of consistency. The alpha level was always set to 0.05, and two-tailed tests were used throughout. We employed Wilcoxon signed-rank tests for comparisons between two conditions and Friedman tests for comparisons between three or more conditions. Data analysis was carried out using SPSS Statistics v. 24.0. For simplicity and consistency with previous studies [15,18], we report mean values throughout the manuscript. All reported tests are planned comparisons based on our *a priori* hypotheses regarding the expected effect of synchronous (illusion) versus asynchronous (control) visuotactile stimulation. Detailed visualizations of the results, including violin plots with means, medians and individual data points, are reported in electronic supplementary material, figure S1.

## 2.8. Experiment 1

The aim of experiment 1 was to examine whether the illusion of owning two full-bodies could be induced through correlated tactile stimulation of the participants' body and visual stimulation of two identical full-bodies viewed from the 1PP (figure 1a). The two bodies used in the visual stimuli were positioned in parallel at an equal distance from the centre of the screen (approx. 5 cm) and are henceforth referred to as 'illusory bodies'. The experiment consisted of four conditions: (i) synchronous stimulation of the real body and both illusory bodies (SS), (ii) asynchronous stimulation of the real body and both illusory bodies (AA), (iii) synchronous stimulation of the real body and the left illusory body but asynchronous stimulation of the right illusory body (SA) and (iv) synchronous stimulation of the real body and the right illusory body but asynchronous stimulation of the left illusory body (AS). The duration of each trial was 2 min, and each condition was repeated once in a pseudorandom trial order that was balanced across participants. After each trial, 10

questionnaire statements (table 1) were presented one-by-one on the screen of the HMDs, and the participants gave a verbal rating for each statement. We hypothesized that synchronized visuo-tactile stimulation would be necessary for ownership and that only the SS condition would be associated with the feeling of owning two bodies simultaneously.

## 2.9. Experiment 2

In experiment 2, we used threat-evoked SCR as a proxy of illusory ownership to test the hypothesis that the synchronous visuo-tactile stimulation of the two illusory bodies would be associated with a greater physiological stress response. Here, after each 1 min period of visuo-tactile stimulation, a hand holding a saw entered the participants' field of view and slid just above the lower legs of one of the illusory bodies for a total duration of 2 s (figure 1b). There were four experimental conditions: (i) SS + left body threat, (ii) SS + right body threat, (iii) AA + left body threat and (iv) AA + right body threat. The trial duration was 1 min, and each condition was repeated three times, yielding a total of 12 trials. The trial order was pseudorandomized and balanced across participants. We predicted that the threat-evoked SCR in the synchronous condition would be greater than in the asynchronous condition for threats directed towards the left and right illusory bodies, respectively.

## 2.10. Experiment 3

Experiment 3 was designed to address the question of whether the threat-evoked SCR findings in experiment 2 were likely to be due to truly simultaneous dual body ownership or, rather, single-body ownership sensations periodically switching between the left and right illusory bodies. To address this issue, we replicated experiment 2 with additional conditions. During the visuo-tactile stimulation phase, we applied either synchronous stimulation to both illusory bodies, synchronous stimulation to the left and asynchronous stimulation to the right, or vice versa. After each of these three stimulation types, the saw threat was applied to either the left or right body, yielding a total of six trial types. For analysis, the trials were collapsed into three conditions: (i) SS + threat either body, (ii) SA/AS + threat synch body and (iii) SA/AS + threat asynch body. We hypothesized that if ownership is indeed switching between the left and right bodies, the threat-evoked SCR magnitude in the SA/AS + threat synch body should be (approx. two times) greater than in the SS + threat either body. This prediction was based on the assumption that in 50% of the SS + threat either body trials, the threatened body should not be owned (because the switching ownership sensations yield a likelihood of 0.5 that participants own the unthreatened body at the time of the threat), which should result in lower SCRs compared with the SA/AS + threat synch body, where the threatened body should be owned in 100% of the trials (because there should be no ownership switching). Conversely, if participants experience true dual ownership, the SCR in the SS + threat either body and SA/AS + threat synch body should not differ significantly. Furthermore, we predicted that both the SS + threat either body and the SA/AS + threat synch body should be associated with a greater SCR than a SA/AS + threat asynch body, in which the 'unowned' body is threatened.

## 2.11. Experiment 4

The goal of experiment 4 was to examine whether the illusion of owning two full-bodies can be used to elicit a sense of dual self-location. Our working hypothesis was that splitting the visual scene vertically as presented in the HMDs from the 1PP in two and inducing ownership of two parallel bodies positioned in distinct spatial environments might create the sense of being in two locations at once. This hypothesis was grounded in the observation that visual information about environmental landmarks from the 1PP contributes to self-location and the known functional links between the senses of body ownership and self-location [15]. To this end, we used the same illusion set-up as in experiments 1–3, with one crucial difference: the 1PP was vertically divided by a thin black line into two equally sized areas, with one of the illusory bodies being located in the testing room and the other illusory body in the corridor outside the testing room (figure 2a). In half of the trials, the left visual hemifield featured a body in the corridor, and in half of the trials, it featured a body in the testing room. The location of the illusory body in the testing room was different from the participant's veridical location (figure 2b). There were two conditions: synchronous (SS) or asynchronous (AA) stimulation of both bodies. The duration of each trial was 2 min, and each condition was repeated once in a pseudorandomized order, balanced across participants. After each trial, the same 10 questionnaire statements used in

experiment 1 were presented one-by-one on the screen of the HMDs, and the participants gave a verbal rating for each statement. Immediately following the questionnaire, the participants were asked to complete a behavioural self-location task. In brief, participants were shown a map over the spatial environment on the HMD screen and were asked to indicate the degree to which they perceived self-location in three labelled candidate positions: the location of the real body in the testing room, the location of the illusory body in the testing room, or the location of the illusory body in the corridor (figure 2b). We hypothesized that the SS would be associated with a stronger sense of dual full-body ownership, in line with the results of experiments 1–3, as well as dual self-location, than AA. The self-location data were analysed by contrasting the two conditions within each of the three candidate locations. We predicted that SS, compared with AA, would be coupled with higher self-location ratings for the two illusory body locations (A and B in figure 2b) and lower self-location ratings for the veridical location (C in figure 2b). In contrast to experiment 5 (see below), participants were not naive to the placement of the beds and cameras in the experiment room.

## 2.12. Experiment 5

The aim of experiment 5 was to explore whether an illusion of dual self-location can be induced by dynamically changing the visual perspective in combination with synchronous visuo-tactile stimulation. We abandoned the approach in experiment 4 of presenting a divided 1PP featuring two full-bodies lying in parallel in separate rooms. Instead, we designed a set-up where two illusory bodies, body A and body B, were lying on two different beds in the same testing room, slightly facing each other. Participants viewed the testing room and the bodies either from the 1PP of body A, the 1PP of body B, or a 3PP in the ceiling above (figure 3a). Body A and B were identical, and the two identical clones of the experimenter delivering touches to the bodies were seen in the periphery of the 3D visual stimulus. To induce a sense of dual self-location, we employed three different types of dynamic changes in visual perspective. Within each 3 min trial, the visual perspective repeatedly 'teleported' every 20 s (duration based on [15]) either between the 1PP of body A and the 1PP of body B (1PP-1PP) or between the two 1PPs and the 3PP in the ceiling (1PP-1PP-3PP), or it remained stationary in the 3PP for the full duration of the trial (3PP only). Our working hypothesis was that 1PP-1PP and 1PP-1PP-3PP, featuring alternating periods of ownership of each of the two bodies viewed from 1PP, could potentially lead to the development of a dual sense of body ownership and self-location. The 3PP-only condition was included to examine whether periods of a 3PP 'bird's eye view' of the room would further facilitate dual self-location, given that this visual perspective provides additional visual information regarding the spatial layout and the dimensions of the environment and the relative distance of the two bodies. Visual stimulation of the two bodies was applied either synchronously (SS) or asynchronously (AA) with respect to the tactile stimulation of the participant, yielding a total of six experimental conditions.

After each trial, the participants were asked to verbally rate questionnaire statements shown in the HMDs. In this experiment, we used a different set of questionnaire statements than in experiments 1 and 4 to better quantify the experience of dual self-location in the current experimental set-up and environment (table 2). After completing the questionnaire, participants were asked to perform a self-location task using the map shown in figure 3b, in which they indicated how strongly they perceived themselves to be located in the positions of body A and B.

It should be noted that great care was taken in leaving the participants naive with respect to their physical location in the experiment room in experiment 5. Before commencing the experimental session, participants familiarized themselves with the testing room, the placement of the beds, and the viewpoints of body A and B for approximately 1 min. They were then led outside, and while they waited there, the experimenter changed the bed placement, making one bed stand between and perpendicular to the original placement of the two beds. This new position did not correspond to any of the two positions of the beds seen in the 3D videos, and the participants were not informed about this change. The participants were then given earplugs and a pair of disconnected HMDs (as a blindfold) and were led back into the testing room. Participants were disoriented by spinning three turns in a standing position and then positioned on the bed, ready to start the experimental session.

Our approach in experiment 5 was exploratory, but nevertheless, we had several working hypotheses based on the previous literature and our experience with full-body illusion paradigms. We hypothesized that SS, compared with AA, would be associated with dual self-location and dual body ownership. In accordance with previous studies showing that 1PP is necessary for full-body ownership [16,38], we predicted that the body ownership illusion would be stronger in the conditions that involved changes in the 1PP (i.e. 1PP-1PP and 1PP-1PP-3PP) compared with the 3PP-only condition. We also expected that

the strongest sense of self-location would be observed in the former two conditions, where participants 'teleport' to each location viewed from the 1PP [15]. The questionnaire data were analysed both with respect to their ratings of individual key dual self-location (S1) and dual full-body ownership statements (S2), as well as the average ratings of all self-location (S1–S3) and all ownership statements (S4–S6) (table 2).

# 3. Results

## 3.1. Experiment 1

Experiment 1 aimed to establish the mere presence of dual body ownership with a questionnaire. As shown in figure 4, synchronous visuo-tactile stimulation of both bodies (SS) was associated with significantly higher ratings of the statements reflecting dual full-body ownership (S1–S2) compared with the asynchronous stimulation of both bodies (AA) (1.4 versus −1.6, $Z = −3.74$, $p < 0.001$), synchronous stimulation of the left but asynchronous stimulation of the right body (SA) (1.4 versus −2.0, $Z = −3.83$, $p < 0.001$), and synchronous stimulation of the right but asynchronous stimulation of the left body (AS) (1.4 versus −1.9, $Z = −3.85$, $p < 0.001$). Notably, the dual-body illusion in SS was driven both by the explicit statement of dual body ownership (S1) and the statement of sensing touches located on both bodies simultaneously (S2), with both statements receiving positive mean rating scores (figure 4). SS was also coupled with significantly higher mean ratings of the statements probing ownership of the left body (S3–S4: 1.1 versus −1.5, $Z = −3.33$, $p = 0.001$) and the right body (S5–S6: 0.9 versus −1.8, $Z = −3.49$, $p < 0.001$) compared with AA. As expected, the conditions SA and AS were coupled with ownership of only the synchronous stimulation body, reflected in the mean ratings of left body ownership (S3–S4) (SA versus AS: 1.9 versus −2.4, $Z = −3.83$, $p < 0.001$) and right body ownership (S5–S6) (AS versus SA: 1.6 versus −2.2, $Z = −3.76$, $p < 0.001$). The average rating of the control statements (S7–S10) did not significantly differ across conditions ($\chi_3^2 = 6.40$, $p = 0.094$). These findings suggest that temporally correlated tactile and visual input from two full-bodies, positioned in parallel and viewed from the 1PP, is associated with simultaneous dual full-body ownership.

## 3.2. Experiments 2 and 3

Experiments 2 and 3 served to corroborate the questionnaire findings of experiment 1. To this end, we examined the threat-evoked SCR associated with the dual full-body ownership condition. Figure 5a shows the results of experiment 2. The threat-evoked SCR magnitude differed significantly across the four conditions ($\chi_3^2 = 15.0$, $p = 0.002$). In line with our prediction, SS, compared with AA, was associated with significantly greater SCR both when the left body (mean $SCR_{ratio}$: 0.21 versus 0.09, $Z = −2.33$, $p = 0.020$) and the right body (mean $SCR_{ratio}$: 0.20 versus 0.06, $Z = −2.90$, $p = 0.004$) were threatened by a saw sliding over the lower legs. These findings suggest that participants experienced ownership over both the left and right illusory bodies in the SS condition.

In further control analyses, we sought to eliminate the potential concern that perhaps the ownership sensations were switching back and forth between the left and the right body rather than being truly simultaneous. To address this question, an experiment quantified the SCR evoked by threats to the left or right bodies in the SS, SA or AS conditions. For analysis, the conditions were collapsed to three: SS + threat body, SA/AS + threat synch body and SA/AS + threat asynch body. If ownership is indeed merely 'switching' between the right and left body, one would expect greater threat-evoked SCR in the SA/AS + threat synch body than in the SS + threat either body because the likelihood of participants owning the threatened body at the time of the threat event should be only 50% in SS, compared with 100% in the SA/AS + threat synch body. The results are shown in figure 5b. The threat-evoked SCR differed significantly across all three conditions ($\chi_2^2 = 7.34$, $p = 0.025$). There was no significant difference between the SS + threat body and the SA/AS + threat synch body (mean $SCR_{ratio}$: 0.25 versus 0.24, $Z = −0.04$, $p = 0.970$), which contradicts the 'ownership switching' hypothesis and supports the notion of simultaneous dual full-body ownership. Finally, both the SS + threat body (mean $SCR_{ratio}$: 0.25 versus 0.13, $Z = −2.11$, $p = 0.035$) and SA/AS + threat synch body (mean $SCR_{ratio}$: 0.24 versus 0.13, $Z = −2.46$, $p = 0.014$) were associated with significantly greater threat-evoked SCR than the SA/AS + threat asynch body. These findings provide additional evidence for successful illusion induction and indicate some degree of spatial specificity of the threat event, since threats toward the (disowned) asynch body evidently did not pose a threat to the owned synch body in the SA/AS + threat asynch body condition.

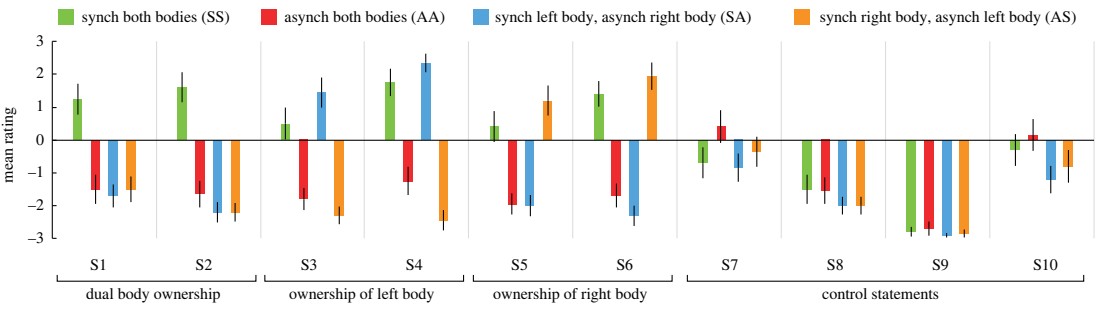

**Figure 4.** Experiment 1: Results. The results of the questionnaire in experiment 1. See table 1 for the questionnaire statements. Error bars represent the standard error.

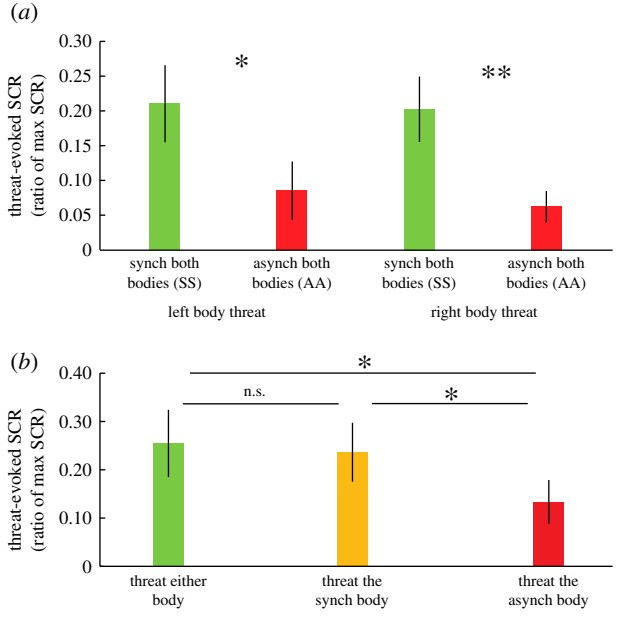

**Figure 5.** Experiments 2 and 3: Results. Threat-evoked SCRs in experiment 2 (*a*) and experiment 3 (*b*). Experiment 2 aimed to establish the presence of a body ownership illusion over two bodies, while experiment 3 served as an additional control on whether dual body ownership is truly simultaneous, or, rather, if single-body ownership transitions between the left and right illusory bodies. Error bars represent the standard error. $^*p < 0.05$, $^{**}p < 0.01$.

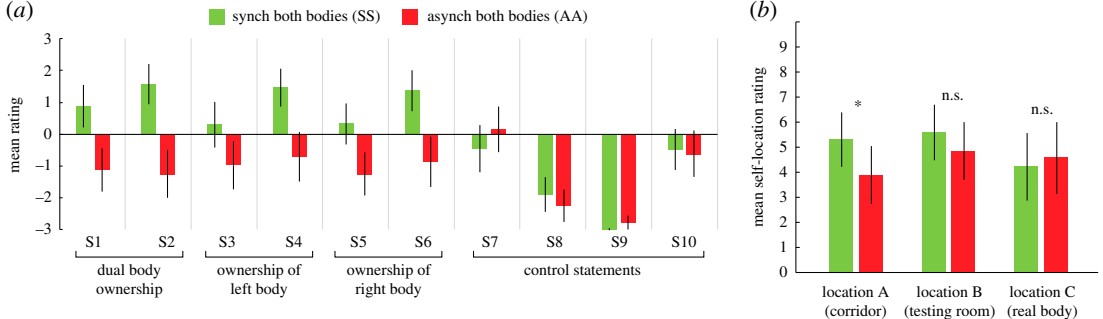

**Figure 6.** Experiment 4: Results. The results of the questionnaire (*a*) and behavioural self-location task (*b*) in experiment 4. See table 1 for questionnaire statements. Error bars represent the standard error. $^*p < 0.05$.

## 3.3. Experiment 4

Experiment 4 explored whether the dual-body ownership illusion combined with a split visual field could lead to an illusion of dual self-location. The results are shown in figure 6a. The condition SS

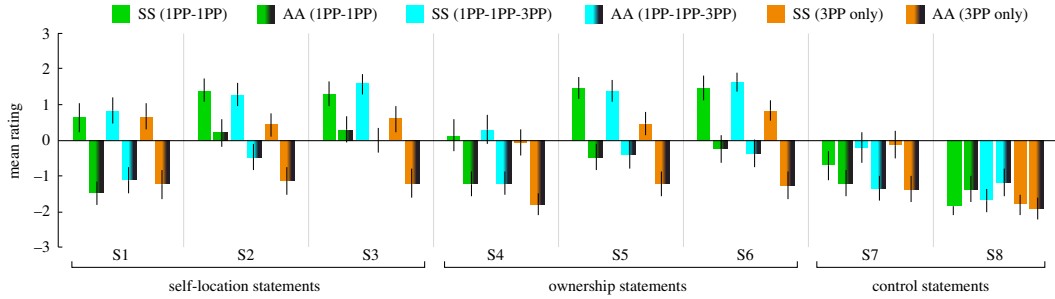

**Figure 7.** Experiment 5: Questionnaire results. See table 2 for the questionnaire statements. Error bars represent the standard error.

was associated with significantly higher mean ratings of the statements reflecting dual full-body ownership (S1–S2) (1.2 versus −1.2, $Z = −3.92$, $p < 0.001$), left body ownership (S3–S4) (0.9 versus −0.8, $Z = −2.91$, $p = 0.004$) and right body ownership (S5–S6) (0.8 versus −1.1, $Z = −3.44$, $p = 0.001$), but there were no differences in the ratings of the control statements (S7–S10) (−1.4 versus −1.4, $Z = −0.42$, $p = 0.674$), compared with AA. Thus, dual full-body ownership was successfully elicited even with a 1PP split in two and the two bodies being located in two different rooms. However, contrary to our hypothesis, dual full-body ownership in SS was not coupled with a robust sense of dual self-location. The results from the self-location task (figure 6b), in which participants rated their sense of self-location in the illusory body in the testing room (location A) versus the illusory body in the corridor (location B) versus the real body location (location C), showed that participants in SS compared with AA rated their self-location slightly stronger (borderline significant) in location A (5.3 versus 3.9, $Z = −1.99$, $p = 0.050$) but there were no differences for location B (5.6 versus 4.8, $Z = −1.09$, $p = 0.28$) or C (4.2 versus 4.6, $Z = −0.83$, $p = 0.40$). These findings suggest that this 'split 1PP' version of the dual full-body illusion cannot be used to manipulate the sense of dual self-location in a predictable manner.

## 3.4. Experiment 5

Experiment 5 examined whether dynamic changes in the visual perspective in combination with synchronous visuo-tactile stimulation can evoke a robust sense of dual self-location. As shown in figure 7, the SS condition was associated with a stronger sense of dual body ownership than AA across the three different visual perspective dynamics. Participants rating the key dual body ownership statement S1 (It felt as if I had two bodies (at the same time)) higher in SS than in AA for both 1PP-1PP (0.1 versus −1.2, $Z = −2.50$, $p = 0.021$), 1PP-1PP-3PP (0.3 versus −1.2, $Z = −3.17$, $p = 0.002$) and 3PP only (−0.1 versus −1.8, $Z = −2.99$, $p = 0.003$) (although it should be noted that the absolute mean ratings in SS were close to zero, suggesting a weak illusion effect). When considering the rating of all ownership-related statements (S4–S6), the SS condition tended to be rated higher in 1PP-1PP (1.0) and 1PP-1PP-3PP (1.1) than in 3PP only (0.4) ($Z = −1.84$, $p = 0.066$; and $Z = −1.94$, $p = 0.052$; with trends toward significance), consistent with the hypothesis that 1PP strengthens the illusion of body ownership and thereby also the dual-body ownership illusion. There was no difference in the ratings of the control statements (S7–S8) when comparing the SS condition with the AA condition (0.0 versus 0.3 versus 0.7, $\chi_2^2 = 2.09$, $p = 0.351$).

It is noteworthy that participants in the SS condition rated the key dual self-location statement S1 (It felt as if I were located in two places at the same time) higher than in AA for 1PP-1PP (0.6 versus −1.5, $Z = −3.63$, $p < 0.001$), 1PP-1PP-3PP (0.8 versus −1.1, $Z = −3.80$, $p < 0.001$) and 3PP only (0.7 versus −1.2, $Z = −3.09$, $p = 0.002$). When considering the mean rating of all self-location-related statements (S1–S3), the SS condition tended to be rated higher in 1PP-1PP (1.1) and 1PP-1PP-3PP (1.2) than in 3PP only (0.6) ($Z = −1.46$, $p = 0.144$; and $Z = −1.79$, $p = 0.074$, with a trend toward significance), suggesting that the absence of 1PP might weaken the illusion of a dual self-location. As shown in figure 8, these questionnaire findings were mirrored in the self-location task results, in which participants in SS compared with AA reported a stronger sense of being in the locations of body A and B (mean rating of A and B) in the 1PP-1PP (6.9 versus 3.9, $Z = −3.86$, $p < 0.001$), 1PP-1PP-3PP (6.7 versus 3.7, $Z = −4.17$, $p < 0.001$) and 3PP-only conditions (5.4 versus 2.6, $Z = −3.75$, $p < 0.001$) (there were no significant differences in the self-location ratings between A and B within each of the six conditions; all pairwise comparisons $p > 0.45$). In SS, the self-location ratings for the conditions involving 1PPs were higher than in the 3PP only (1PP-1PP versus 3PP only: 6.9 versus 5.4, $Z = −3.28$,

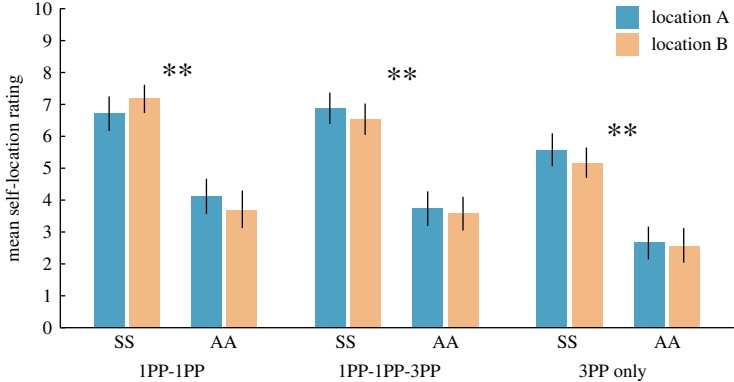

**Figure 8.** Experiment 5: Self-location task results. $**p < 0.01$. Error bars represent the standard error.

$p = 0.001$; and 1PP-1PP-3PP versus 3PP only: 6.7 versus 5.4, $Z = -2.70$, $p = 0.007$). In summary, these findings seem to suggest that dynamic changes in the visual perspective in combination with synchronous visuo-tactile stimulation can manipulate the senses of dual full-body ownership and dual self-location in a predictable manner. Furthermore, involving the 1PP appears to boost the illusion both in terms of dual self-location and dual body ownership.

## 4. Discussion

We used a full-body ownership illusion [18] and manipulations of the visual perspective, illusory body placement in the local environment, and visuo-tactile synchrony to investigate whether this illusion can be used to induce the perception of owning two distinct full-bodies and being in two separate spatial locations at the same time. In support of our first hypothesis, we found that synchronous visuo-tactile stimulation of the participants' body and two virtual bodies viewed from the 1PP, placed in close proximity side-by-side, elicited a strong sense of dual full-body ownership, supported by the questionnaire and threat-evoked SCR measurements (experiments 1–3). This finding extends earlier work on supernumerary limb illusions to the case of experiencing ownership over two entire bodies viewed side-by-side from the 1PP and suggests that the perceived oneness of the bodily self is the result of active multisensory integration processes. However, using the illusion to elicit an experience of dual self-location for artificial bodies placed far apart, our second exploratory aim, turned out to be a more challenging task. Our initial hypothesis that dual self-location would be induced by splitting the visual field of the 1PP into two equally sized halves (left and right) and placing each of the owned illusory bodies in separate spatial environments was falsified in experiment 4. In the final exploratory experiment 5, we found that dynamic changes in the visual perspective from 1PP to 3PP combined with 'illusory teleportation' between two locations in a testing room led to an apparent illusion of dual self-location as well as dual self-ownership, as indicated by the questionnaires. Together, these findings suggest that multisensory integrative mechanisms can be used to manipulate the senses of dual full-body ownership and dual self-location in a predictable manner, which supports multisensory theories of bodily self [35,39,40] and self-consciousness [27,41], and opens up new avenues of empirical research into the perceived unity and oneness of bodily self.

The dual full-body ownership illusion reported in experiments 1–3 was predicted based on previous studies on illusory duplications of single limbs [11–14]. The possibility of the brain to accept extra illusory limbs may not be overly surprising, given the substantial evidence for supernumerary phantom limbs occurring following brain lesions [1,42–46]. Patients reporting the experience of full-body phantoms, so-called he-autoscopy, are more rare and have been associated with strikingly different conditions (epilepsy [47], astrocytoma in the insular region of the left temporal lobe [48], schizophrenia [49], or major depression [50]). Some he-autoscopic experiences are described as a phantom duplication of the patient's body from the neck and down. Although we did not explicitly evaluate whether our participants experienced having one or two heads during the illusion, no one spontaneously reported having two separate heads, leading us to assume that illusory body duplication was restricted to the neck/chest and down (as shown in the visual stimulus, figure 1a). The present ownership illusion is also distinct from out-of-body experiences (OBEs) of neurological origin, which always involve a spatial separation of the head of the patient and the head of the illusory body [5,6,8]. In light of this,

we propose that the existence of the present dual full-body ownership illusion represents a substantial extension of previous work because it demonstrates that multisensory integration mechanisms in the healthy brain can lead to the unique perception of having two distinct full-bodies from the neck/chest and down.

The present results go beyond those of the study by Heydrich *et al.* [20]. In that study, participants self-identified with two virtual bodies viewed from a distance when synchronous strokes were applied to the participant's back and the two virtual bodies' backs. However, the supernumerary body effect was phenomenologically weak with low subjective ratings of key questionnaire items (see Introduction), which is consistent with previous studies showing that full-body illusion paradigms where the body is viewed from a 3PP produce weaker ownership compared with paradigms using a 1PP [16,38,51,52]. In the present experiment 1, participants in the SS condition clearly agreed to statements reflecting dual-body ownership (S1) and referral of touch to both bodies in view (S2), which is in line with the results from previous supernumerary limb illusions [12,13] but here involving two entire bodies from the neck down. In contrast to [20], these questionnaire results were corroborated by the threat-evoked SCR results in experiments 2 and 3, providing physiological evidence that both bodies were represented as belonging to self in the SS condition.

What could be the mechanisms of the dual-body ownership illusion? How does this illusion fit into current theories of body representation? Body ownership illusions are intimately associated with multisensory integration [53] and are thought to arise as a consequence of the brain's normal machinery for combining signals across sensory modalities to produce a flexible and accurate perception of the body in space. This multisensory integration process is dynamic and influenced by previous experience (and innate factors) and the spatio-temporal patterns of afferent sensory signals from the different modalities and their relative reliability (the signal-to-noise ratio) [54–56]. Thus, an illusory body ownership percept arises when visual, tactile and proprioceptive signals are combined into coherent multisensory representation of one's own body that is inconsistent with the physical reality [35,39,57–60]. The present illusion of owning two bodies at the same time reveals a surprising flexibility in this dynamic multisensory process of own-body perception: the visual input from the two bodies being touched is perceptually fused with the tactile, proprioceptive, and other somatic signals from the hidden real body into two coherent multisensory representations of the bodies (from the torso down). The existence of this dual-body ownership illusion suggests that the perceived oneness of the bodily self is not as static as one might think. Rather than deriving the numerosity of one's body based on the ego-centric spatial reference frame or from extremely strong priors related to previous experiences and/or innate factors, the present findings suggest that perceived body numerosity is the result of a dynamic multisensory integration process that continuously infers the most likely spatial model of the own-body. If the available sensory signals are truly ambiguous with regard to body numerosity, this internal model may accept the conceptually impossible scenario of having two bodies simultaneously. The current illusion thus generalizes earlier results regarding supernumerary limb illusions to the case of the entire body. The flexibility in terms of binding vision, touch and proprioception into one single or multiple multisensory own-body percepts does not seem to be restricted to the case of the upper limb, which might be assumed given the mobility of the arm and hand but also applies to immobile core regions of the body, such as the torso and abdomen. This notion indicates, speculatively, that the same probabilistic principles for multisensory combination and causal inference that govern the localization of external objects [61–63] and limbs in space [58,64,65] might determine the perception of the bodily self as a whole.

In the second part of the study, we sought to explore alternative ways of creating the illusion of dual self-location and dual ownership by manipulating the visual perspective and visual scene. Our first attempt in this direction, placing the two illusory bodies in distinct spatial environments by splitting the 1PP into two, resulted in dual full-body ownership but, contrary to our hypothesis, did not lead to a dual self-location (experiment 4). Our working hypothesis was that by splitting the 1PP in two and placing the two owned bodies in the two different environments, the testing room and the corridor outside the testing room, the brain would receive ambiguous and equally strong evidence for self-location in two distinct rooms, resulting in a feeling of being located in the two places simultaneously. After testing, some participants spontaneously reported that they indeed experienced having two bodies but that the room in which the illusory bodies were located looked 'weird'. Our interpretation of these reports and our own impressions after testing the paradigm ourselves is that the two visual scenes perceptually fused into a single (weird-looking) environment, rather than the bodily self-location being split between two distinct spatial representations. This interpretation was mirrored in the self-location task results in which no difference in self-location rating was observed

between the ownership (synchronous) and non-illusion (asynchronous) conditions. These findings suggest that the experience of a unitary spatial origin of visual 1PP is resistant to correlated multisensory cues, suggesting dual self-location. Whereas ambiguous multisensory stimulation can make the brain accept physically inconsistent ownership sensations, such as having two right arms [13], a slowly growing very long arm [66], an invisible body [34], or two full-bodies, providing the brain with similar ambiguous input regarding the spatial origin of the 1PP simply leads to the perception of being in a single room that looks divided and strange. We speculate that having one single spatial origin of the 1PP is so fundamental that it is more resilient to ambiguous multisensory input than the process of localizing the body in space, which arguably involves a higher degree of noise and uncertainty.

The failed attempt to induce dual self-location in experiment 4 prompted us to try another explorative approach in experiment 5. Instead of using a 'split' 1PP with two bodies positioned in parallel but in different rooms, we dynamically changed the visual perspective every 20 s between separate 1PPs of two illusory bodies placed in different corners of the testing room and a common 3PP in the ceiling looking down at both bodies. Whereas the two illusory bodies in experiment 4 were placed close to each other in parallel, within the theoretical limits of each other's peripersonal space, the two illusory bodies in experiment 5 were spatially separated far apart, well beyond each other's peripersonal space. The results showed that the dynamic changes of the visual perspective in combination with synchronous visuo-tactile stimulation was associated with a feeling of dual full-body ownership and dual self-location. Our interpretation is that the repeated ownership experience of each body viewed from 1PP, in combination with always seeing 'the other body' from afar while receiving identical synchronous visuotactile stimulation, resulted in an accumulation of spatial experience of being located at both places at the same time, and both bodies were equally owned. An alternative interpretation is that the participants only perceived genuine body ownership and self-location for one body at a time—the body currently viewed from the 1PP—and that the dual ownership and self-location reported in the questionnaire resulted from a post-perceptual explicit reflection of the overall experience. Future experiments are needed to conclusively address these two alternative interpretations by repeatedly probing ownership and self-location during different phases of the experience.

The condition in which participants in experiment 5 observed the two illusory bodies from one single 3PP in the ceiling also appeared to be coupled with dual full-body ownership and dual self-location. However, the strength of the illusion in the 3PP-only condition tended to be weaker than that in the 1PP-1PP and 1PP-1PP-3PP conditions in terms of questionnaire ratings. These results are in line with previous work on the (single) full-body illusion showing that the 1PP of the illusory body is important for ownership [16,17,38]. It should be noted that there are several limitations of experiment 5, including the lack of objective behavioural evidence for the ownership illusion (SCR) or a control for truly simultaneous ownership and self-location versus switching back and forth between the two illusory bodies. The results from experiments 4 and 5 should therefore be considered preliminary and interpreted with caution and should be viewed as hypothesis-generating for future studies examining the minimal conditions required for eliciting dual self-location. Moreover, in all five experiments, participants were merely passively experiencing an illusion; they had no ability to manipulate the virtual bodies in space, and we did not explicitly control their focus of attention. Future studies should investigate to what extent dual ownership allows for independent control of two bodies at once and whether the illusion is constrained by limited resources of spatial attention (as measured by, for instance, eye tracking).

Finally, it is interesting to consider the possible neural mechanisms that could potentially underlie simultaneous dual full-body ownership and dual self-location. We hypothesized that ownership of two bodies could be implemented by populations of multisensory neurons in the premotor and posterior parietal cortices, which have been implicated in previous fMRI studies of full-body ownership illusions with single bodies [15,17,67]. Speculatively, subpopulations of cells in these areas might be implementing the integration of somatosensory and visual information from each of the two bodies separately, and activity in the premotor cortex might differentiate between a (causal inference) model [57,58] involving one or two own-bodies. Regarding dual self-location, we speculate that a circuit involving the hippocampus, retrosplenial, posterior cingulate and posterior parietal cortices might play a critical role [15]. We speculate that different groups of place cells [68] might simultaneously be active to represent the locations of each of the two bodies at the same time. This contrasts with the more common idea that hippocampal place cells are very selective and are involved in differentiating instead of integrating overlapping spatial contexts [69]. Thus, the present dual-body illusion could be used to test fundamental assumptions regarding place cells and self-location

representation. Based on the above, we predict that it should be possible to read out dual- versus single-body ownership and dual versus single self-location by analysing fine-grained neural activity patterns in the premotor-intraparietal and parieto-cingulate-hippocampal areas, respectively.

In summary, the findings of this study suggest that two basic aspects of bodily self-perception, the feelings of having a single body and being in one spatial location, are more malleable than commonly assumed. The continued use of supernumerary full-body illusions may be a powerful tool for probing the limits of multisensory and spatial body representations and for examining the mechanisms underlying the perceived unity between body ownership and self-location. Historically, illusions have been important for psychology and neuroscience because of what they can tell us about how perception works under normal conditions. From this perspective, the present illusion is valuable because it could inform us about the processes that support one of the most fundamental aspects of the human subjective experience: the perceived oneness of the bodily self.

Ethics. All subjects gave their written informed consent prior to participation, and the Regional Ethical Review Board of Stockholm approved all of the experimental procedures. The experiments were performed in accordance with relevant guidelines and regulations.

Data accessibility. Source data are available in electronic supplementary material.

Authors' contributions. A.G. and H.H.E. designed research; D.E.O.L. and J.S. performed research, A.G. analysed data; A.G. drafted the paper; D.E.O.L., J.S. and H.H.E. provided critical revisions.

Competing interests. We declare we have no competing interests.

Funding. J.S. received support from the Polish Ministry of Science and Higher Education "Diamond Grant".

Acknowledgments. This work was supported by the Swedish Research Council and the European Research Council (SELF-UNITY). A.G. was supported by the Wenner-Gren Foundation, the Sweden-America Foundation and the Promobilia Foundation.

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
