## [Reviewer comments · Royal Society Open Science]

Review History

RSOS-200361.R0 (Original submission)

Review form: Reviewer 1

Is the manuscript scientifically sound in its present form?

Yes

Are the interpretations and conclusions justified by the results?

No

Is the language acceptable?

Yes

Do you have any ethical concerns with this paper?

No

Have you any concerns about statistical analyses in this paper?

No

Recommendation?

Reject

Comments to the Author(s)

This is an extensive exploration of the possibility of embodying more than one body by using multisensory manipulations. 5 Experiments are presented, testing the embodiment of 2 virtual body close to each other (exp 1-2), measuring both subjective (questionnaires) and physiological (GSR) correlates of the effects, and two bodies that are virtually presented in different location (exp 3,4). In this case a more novel question is asked, about whether embodying multiple bodies, occupying different spatial locations, would result in multiple self-locations.

The experiments are sound and methodologically correct.
I am wondering about the real novelty about the current results.

The results from experiment 1-3 better support and complement previous findings by Aymerich-Franch et al. and Heydrich et al. - and support the conclusion that people can feel ownership for more than one virtual body, under specific conditions.

However, the most novel and interesting question is whether people can genuinely experience to be in 2 locations at the same time. This was really tested in Experiment 5, as in experiment 4, the two bodies were anyway shown close to each other, although in split halves of the screen (but still potentially perceived within one's own PPS).

Unfortunately, results from Experiment 5 are not conclusive. Questionnaire scores reported higher ratings for self-location items in the synchronous vs. asynchronous conditions, but ratings were generally low (around 1). Now, these illusions are normally mild, and it is normally to not find very higher scores, but unfortunately this single very mild effect is the only one supporting the hypothesis of a shift in self-location. Indeed, results from the self-location task are not conclusive: the task is performed at the end of a 3-minutes long stimulation alternating between the two point of view. It is possible that the participants report high scores both for location A and for location B by alternating from one to another, as a function of the stimulation, without never experiencing being in two places at the same time.

Self location results (5B). There is something surprisingly in the self-location ratings. One should expect higher self location at location of the physical body, at least in the asynchronous conditions. Then eventually these ratings should increase for the two avatars' location in the synchronous condition. Instead, the ratings are relatively low for the physical body. Does this depend on just showing a different point of view from the HDM? I am wondering whether the authors collected a "baseline" rating, with no HMD, or HMD on but no body being shown, to make sense of this relatively lower self-location within the physical body.

Minor:

The paper contains 5 experiments, whose rationale is explained in a long method section, pp. 10-14, which presents in details each experiment.

When reading the results, the reader has to go back to this section to properly understand what the authors are testing in each result section. I suggest to add to each results section a brief paradigm to remind the reader the key hypothesis tested / manipulation for that specific experiment.

Results, Experiment 2-3 pp. 14-15 : Add a comment / explanation to Figure 3B, to better link to Experiment 3.

Page 17-18:

"However, to the best of the authors' knowledge, there are no known cases of brain-lesion induced full-body phantoms, defined as a phantom duplication of the patient's body from the neck and down".

This is actually the definition of "he-autoscopy" (see e.g., Brugger, P., Agosti, R., Regard, M., Wieser, H., and Landis, T. (1994). Heautoscopy, epilepsy and suicide. *J. Neurol. Neurosurg.*

Psychiatry 57, 838-839.). There is no single clear brain lesion resulting in this phenomenon, but for sure it has been reported.

Review form: Reviewer 2

Is the manuscript scientifically sound in its present form?

Yes

Are the interpretations and conclusions justified by the results?

Yes

Is the language acceptable?

Yes

Do you have any ethical concerns with this paper?

No

Have you any concerns about statistical analyses in this paper?

No

Recommendation?

Accept with minor revision (please list in comments)

Comments to the Author(s)

This manuscript investigates whether the experience of dual body-ownership and self-location can be induced through sensory stimulation and perceptual illusions. In a natural development from previous virtual reality studies, the experiment deploys comprehensive methods to probe the mechanisms that account for a supernumerary self. The techniques are solid and the results are easy to understand. Despite that, there are a few points I'd like to address:

1 - Is there a reason for the number of participants chosen (pp 6 line 49) based on the referenced study?

2 - On page 6, "Visual stimuli and head mounted displays", there is no mention of the focal length of the video the subject watched, the perceived angle of view or the display's field of view. It would also be interesting to clarify if head movement had any effect on the video (and to what degree) or if the video was fixed position-wise. Since the video contains objects moving relatively fast in a fixed background (the tactile stimuli) the frame rate and display refresh rate used during the experiment should also be mentioned. These factors influence the feeling of immersion and presence, which could affect the sense of body ownership. Although small artifacts or a low refresh rate may not hinder single body illusions, the decrease in immersion could affect more complex illusions (such as dual body); which could also account for different results in other experiments. Besides, this information is essential to replicate the study.

3 - Page 9 line 8 refers to "Fig. 1A", but figure 1 has no clear subdivisions.

4 - Page 9 lines 15-21: both conditions described are the same, although the acronyms (SA, AS) are correct: "(3) synchronous stimulation the real body and the left illusory body but asynchronous stimulation right illusory body (SA), (4) synchronous stimulation the real body and the left illusory body but asynchronous stimulation right illusory body (AS)."

5- Page 10 lines 15-28: Could attention affect the likeness of the body being owned? The synchronized stimuli may draw attention to one body over the other and attention is a process

intertwined with conscious perception. When both bodies were displayed at the same time, eye recording would be able to tell if attention to each body was asymmetric (and dependent on stimulus synchronization).

6 - Page 12 lines 26,27: were the participants naive to their real position only on experiment 5? Was the lack of significant difference in the results between left and right bodies enough to affirm that knowing their real position had no effect in the illusion in the other experiments?

7 - Page 14 line 19: I assume the figure called is 3B instead of 4B.

8 - Page 15 line 15: Statement S4, not S5.

9 - Page 15: lines 12-19: The affirmation that "the SS condition was associated with a stronger sense of dual body ownership than AA across the three different visual perspective dynamics" is misleading. Although participants indeed rated higher dual body ownership questions in SS over AA conditions, the results show numbers close to 0 (0.1, 0.3 and -0.1 respectively, lines 17-19), which indicated "I do not know, I can neither agree nor disagree" on the scale (page 7 line 22). This could be clarified in the text. I also wonder if the participants relied on the scale's visual proportions instead of the semantics of the score.

10 - Page 16 lines 33-38: Confusing sentence: "Our initial working hypothesis that splitting the visual field of the 1PP into two equally sized halves (left and right) and placing each of the owned illusory bodies in separate spatial environments was falsified in experiment 4.". Which working hypothesis was falsified, experiment 4 (dual self location in distinct environments)?

11 - Page 20 lines 21-22: "We speculate that different groups of place cells [62] might simultaneously be active to represent the locations of each of the two bodies at the same time.". This is an interesting hypothesis that conflicts with the more common idea that hippocampal place cells are very selective and involved in differentiating instead of integrating overlapping spatial contexts. References with experimental evidence may be necessary to improve the credibility of this speculation.

12 - Figure 7: On the graph legend, I suggest avoiding mixing + and - symbols when these have meanings other than addition and subtraction. Since - is already used on the text, maybe changing + to another symbol may make the graph clearer without compromising the concordance with the main text. For example: SS(1PP-1PP) instead of SS+1PP-1PP.

13 - The figures lack a description of what is being shown. For example, the main hypothesis of the respective experiment, the purpose of what is being shown, or the main conclusion the figures present.

14 - The bottom legend on figures 2, 5A, and 7 about the class of statements could have a more clear divider. For example, in figure 2, without checking the statements table, it is difficult to tell if "control statements" are only S8,S9 or S7-S10.

- Grammar:

Page 9 line 29 (only the SS condition was be associated with the)

page 11 line 10 (the participants gave a verbal rating of for each statement)

Review form: Reviewer 3

Is the manuscript scientifically sound in its present form?

Yes

Are the interpretations and conclusions justified by the results?

Yes

Is the language acceptable?

Yes

Do you have any ethical concerns with this paper?

No

Have you any concerns about statistical analyses in this paper?

Yes

Recommendation?

Accept with minor revision (please list in comments)

Comments to the Author(s)

The paper reports several multisensory experiments, aiming to produce the experience of having two bodies. The authors conclude this experience can be induced.

Strengths of the study include the number of participants and number of studies. A weakness lies in the way experience is measured. Questionnaire measures of illusions may involve an element of suggestion, and can induce task demands that experimenters should ideally avoid. Threat responses are informative, but do not provide much information about underlying mechanism. The self-location measures here seem to be responses to questionnaire items, rather than direct judgements about the agent's location within a field. Thus, when participants apparently report being in two places at once, it is unclear whether this is linked to the way the question is asked (people are often inconsistent in reply to questionnaire items where one might expect consistency, e.g., "How anxious are you?"/"How worried are you?").

In expts 1-3, the participant views two virtual sticks stroking two different virtual legs. The viewpoint seems to be shifted towards the middle of the two visual bodies. One virtual body has their left leg stimulated, and the other virtual body as their right leg stimulated. However, it is unclear whether tactile stimulation is applied to both legs - this should be in the Methods.

Decision letter (RSOS-200361.R0)

Dear Dr Guterstam:

Manuscript ID RSOS-200361 entitled "Duplication of bodily self: a perceptual illusion of dual full-body ownership and dual self-location" which you submitted to Royal Society Open Science, has been reviewed. The comments from reviewers are included at the bottom of this letter.

In view of the criticisms of the reviewers, the manuscript has been rejected in its current form. However, a new manuscript may be submitted which takes into consideration these comments.

Please note that resubmitting your manuscript does not guarantee eventual acceptance, and that your resubmission will be subject to peer review before a decision is made.

Your resubmitted manuscript should be submitted by 26-Nov-2020. If you are unable to submit by this date please contact the Editorial Office.

on behalf of Dr Atsushi Iriki (Associate Editor)
openscience@royalsociety.org

Reviewers' Comments to Author:

Reviewer: 1
Comments to the Author(s)

This is an extensive exploration of the possibility of embodying more than one body by using multisensory manipulations. 5 Experiments are presented, testing the embodiment of 2 virtual body close to each other (exp 1-2), measuring both subjective (questionnaires) and physiological (GSR) correlates of the effects, and two bodies that are virtually presented in different location (exp 3,4). In this case a more novel question is asked, about whether embodying multiple bodies, occupying different spatial locations, would result in multiple self-locations.

The experiments are sound and methodologically correct.
I am wondering about the real novelty about the current results.

The results from experiment 1-3 better support and complement previous findings by Aymerich-Franch et al. and Heydrich et al. - and support the conclusion that people can feel ownership for more than one virtual body, under specific conditions.

However, the most novel and interesting question is whether people can genuinely experience to be in 2 locations at the same time. This was really tested in Experiment 5, as in experiment 4, the two bodies were anyway shown close to each other, although in split halves of the screen (but still potentially perceived within one's own PPS).

Unfortunately, results from Experiment 5 are not conclusive. Questionnaire scores reported higher ratings for self-location items in the synchronous vs. asynchronous conditions, but ratings were generally low (around 1). Now, these illusions are normally mild, and it is normally to not find very higher scores, but unfortunately this single very mild effect is the only one supporting the hypothesis of a shift in self-location. Indeed, results from the self-location task are not conclusive: the task is performed at the end of a 3-minutes long stimulation alternating between the two point of view. It is possible that the participants report high scores both for location A and for location B by alternating from one to another, as a function of the stimulation, without never experiencing being in two places at the same time.

Self location results (5B). There is something surprisingly in the self-location ratings. One should expect higher self location at location of the physical body, at least in the asynchronous conditions. Then eventually these ratings should increase for the two avatars' location in the synchronous condition. Instead, the ratings are relatively low for the physical body. Does this depend on just showing a different point of view from the HDM? I am wondering whether the authors collected a "baseline" rating, with no HMD, or HMD on but no body being shown, to make sense of this relatively lower self-location within the physical body.

Minor:

The paper contains 5 experiments, whose rationale is explained in a long method section, pp. 10-14, which presents in details each experiment.

When reading the results, the reader has to go back to this section to properly understand what the authors are testing in each result section. I suggest to add to each results section a brief paradigm to remind the reader the key hypothesis tested / manipulation for that specific experiment.

Results, Experiment 2-3 pp. 14-15 : Add a comment / explanation to Figure 3B, to better link to Experiment 3.

Page 17-18:

"However, to the best of the authors' knowledge, there are no known cases of brain-lesion induced full-body phantoms, defined as a phantom duplication of the patient's body from the neck and down".

This is actually the definition of "he-autoscopy" (see e.g., Brugger, P., Agosti, R., Regard, M., Wieser, H., and Landis, T. (1994). Heautoscopy, epilepsy and suicide. *J. Neurol. Neurosurg. Psychiatry* 57, 838-839.). There is no single clear brain lesion resulting in this phenomenon, but for sure it has been reported.

Reviewer: 2

Comments to the Author(s)

This manuscript investigates whether the experience of dual body-ownership and self-location can be induced through sensory stimulation and perceptual illusions. In a natural development from previous virtual reality studies, the experiment deploys comprehensive methods to probe the mechanisms that account for a supernumerary self. The techniques are solid and the results are easy to understand. Despite that, there are a few points I'd like to address:

1 - Is there a reason for the number of participants chosen (pp 6 line 49) based on the referenced study?

2 - On page 6, "Visual stimuli and head mounted displays", there is no mention of the focal length of the video the subject watched, the perceived angle of view or the display's field of view. It would also be interesting to clarify if head movement had any effect on the video (and to what degree) or if the video was fixed position-wise. Since the video contains objects moving relatively fast in a fixed background (the tactile stimuli) the frame rate and display refresh rate used during the experiment should also be mentioned. These factors influence the feeling of immersion and presence, which could affect the sense of body ownership. Although small artifacts or a low refresh rate may not hinder single body illusions, the decrease in immersion could affect more complex illusions (such as dual body); which could also account for different results in other experiments. Besides, this information is essential to replicate the study.

3 - Page 9 line 8 refers to "Fig. 1A", but figure 1 has no clear subdivisions.

4 - Page 9 lines 15-21: both conditions described are the same, although the acronyms (SA, AS) are correct: "(3) synchronous stimulation the real body and the left illusory body but asynchronous stimulation right illusory body (SA), (4) synchronous stimulation the real body and the left illusory body but asynchronous stimulation right illusory body (AS)."

5- Page 10 lines 15-28: Could attention affect the likeness of the body being owned? The synchronized stimuli may draw attention to one body over the other and attention is a process intertwined with conscious perception. When both bodies were displayed at the same time, eye recording would be able to tell if attention to each body was asymmetric (and dependent on stimulus synchronization).

6 - Page 12 lines 26,27: were the participants naive to their real position only on experiment 5? Was the lack of significant difference in the results between left and right bodies enough to affirm that knowing their real position had no effect in the illusion in the other experiments?

7 - Page 14 line 19: I assume the figure called is 3B instead of 4B.

8 - Page 15 line 15: Statement S4, not S5.

9 - Page 15: lines 12-19: The affirmation that "the SS condition was associated with a stronger sense of dual body ownership than AA across the three different visual perspective dynamics" is misleading. Although participants indeed rated higher dual body ownership questions in SS over AA conditions, the results show numbers close to 0 (0.1, 0.3 and -0.1 respectively, lines 17-19), which indicated "I do not know, I can neither agree nor disagree" on the scale (page 7 line 22). This could be clarified in the text. I also wonder if the participants relied on the scale's visual proportions instead of the semantics of the score.

10 - Page 16 lines 33-38: Confusing sentence: "Our initial working hypothesis that splitting the visual field of the 1PP into two equally sized halves (left and right) and placing each of the owned illusory bodies in separate spatial environments was falsified in experiment 4.". Which working hypothesis was falsified, experiment 4 (dual self location in distinct environments)?

11 - Page 20 lines 21-22: "We speculate that different groups of place cells [62] might simultaneously be active to represent the locations of each of the two bodies at the same time.". This is an interesting hypothesis that conflicts with the more common idea that hippocampal place cells are very selective and involved in differentiating instead of integrating overlapping spatial contexts. References with experimental evidence may be necessary to improve the credibility of this speculation.

12 - Figure 7: On the graph legend, I suggest avoiding mixing + and - symbols when these have meanings other than addition and subtraction. Since - is already used on the text, maybe changing + to another symbol may make the graph clearer without compromising the concordance with the main text. For example: SS(1PP-1PP) instead of SS+1PP-1PP.

13 - The figures lack a description of what is being shown. For example, the main hypothesis of the respective experiment, the purpose of what is being shown, or the main conclusion the figures present.

14 - The bottom legend on figures 2, 5A, and 7 about the class of statements could have a more clear divider. For example, in figure 2, without checking the statements table, it is difficult to tell if "control statements" are only S8,S9 or S7-S10.

- Grammar:

Page 9 line 29 (only the SS condition was be associated with the)
page 11 line 10 (the participants gave a verbal rating of for each statement)

Reviewer: 3
Comments to the Author(s)

The paper reports several multisensory experiments, aiming to produce the experience of having two bodies. The authors conclude this experience can be induced.

Strengths of the study include the number of participants and number of studies. A weakness lies in the way experience is measured. Questionnaire measures of illusions may involve an element of suggestion, and can induce task demands that experimenters should ideally avoid. Threat responses are informative, but do not provide much information about underlying mechanism. The self-location measures here seem to be responses to questionnaire items, rather than direct judgements about the agent's location within a field. Thus, when participants apparently report being in two places at once, it is unclear whether this is linked to the way the question is asked (people are often inconsistent in reply to questionnaire items where one might expect consistency, e.g., "How anxious are you?"/"How worried are you?").

In expts 1-3, the participant views two virtual sticks stroking two different virtual legs. The viewpoint seems to be shifted towards the middle of the two visual bodies. One virtual body has their left leg stimulated, and the other virtual body as their right leg stimulated. However, it is unclear whether tactile stimulation is applied to both legs - this should be in the Methods.

Author's Response to Decision Letter for (RSOS-200361.R0)

See Appendix A.

Decision letter (RSOS-201911.R0)

Dear Dr Guterstam,

I am pleased to inform you that your manuscript entitled "Duplication of bodily self: a perceptual illusion of dual full-body ownership and dual self-location" is now accepted for publication in Royal Society Open Science.

Royal Society Open Science operates under a continuous publication model. Your article will be published as soon as it is ready for publication, and this will be the final version of the paper. As such, it can be cited immediately by other researchers. As the issue version of your paper will be the only version to be published I would advise you to check your proofs thoroughly as changes cannot be made once the paper is published.

Articles are normally press released. For this to be effective we set an embargo on news coverage corresponding to the publication date of the article. We request that news media and the authors do not publish stories ahead of this embargo (when final version of the article is available). Please see the Royal Society Publishing guidance on how you may share your accepted author manuscript at <https://royalsociety.org/journals/ethics-policies/media-embargo/>.

on behalf of Dr Atsushi Iriki (Associate Editor) and the Subject Editor
openscience@royalsociety.org

Appendix A

Point-by-point response to reviewers

We thank the three reviewers for their thoughtful, careful and detailed comments. We have revised the manuscript to take these comments into account, and we think the paper is greatly improved. Below we answer all comments in a point-by-point manner.

Reviewer 1

This is an extensive exploration of the possibility of embodying more than one body by using multisensory manipulations. 5 Experiments are presented, testing the embodiment of 2 virtual body close to each other (exp 1-2), measuring both subjective (questionnaires) and physiological (GSR) correlates of the effects, and two bodies that are virtually presented in different location (exp 3,4). In this case a more novel question is asked, about whether embodying multiple bodies, occupying different spatial locations, would result in multiple self-locations. The experiments are sound and methodologically correct.

Authors' response:

We are very pleased to read the reviewer's general positive opinion of our study, and we would like to thank him or her for the insightful comments provided. We hope that our point-by-point responses below will satisfactorily address the issues raised.

Point #1

I am wondering about the real novelty about the current results.

Authors' response:

We are grateful to the reviewer's comment, because it has helped us to clarify the advance with respect to the previous literature. To our knowledge, dual full-body ownership and dual self-location have not been thoroughly investigated as only two experiments have previously attempted to induce it. Notably, previous studies significantly differ in their methodological approaches compared to ours. In Heydrich et al. (2013) virtual bodies were presented from the distance (third person perspective), neglecting the importance of the first person perspective in triggering body ownership illusions (Petkova, Khoshnevis & Ehrsson, 2011). This could explain the authors' failure to induce dual body ownership over two bodies simultaneously as assessed by the authors' questionnaires. In Aymerich-Franch et al (2016), the questionnaire did not evaluate the degree of ownership of the seen real body and contained no statements to control for suggestibility or task compliance, making the results difficult to interpret. In addition, that study did not quantify ownership and self-location with any indirect behavioral or physiological measurement. This renders our study the first successful and methodologically robust attempt to elicit dual body ownership illusion in healthy participants. In the revised discussion, we have made several

clarifications and also added a new paragraph (see below) to clarify the conceptual advance and methodological differences between the current study and the results presented in Heydrich et al. 2013. We hope the reviewer and editor agree that this work constitutes a significant advance of knowledge.

Discussion, paragraph 3:

“The present results go beyond those of Heydrich et al. (2013). In that study, participants self-identified with two virtual bodies viewed from a distance when synchronous strokes were applied to the participant’s back and the two virtual bodies’ backs. However, the effect was phenomenologically weak with low subjective ratings (see Introduction), which is consistent with previous studies showing that full-body illusion paradigms where the body is viewed from a 3PP produce weaker ownership compared to paradigms using the 1PP [16,37,68-69]. In the present experiment 1, participants gave affirmative ratings for both subjective dual-body ownership and duplication of referred touch to both alien bodies in view, which is in line with to the results from previous supernumerary limb illusions [12,13] but here involving two entire bodies from the neck down. In contrast to [8], these questionnaire results were corroborated by the threat-evoked SCR results in experiments 2 and 3, providing physiological evidence that both bodies were represented as belonging to self in the SS condition.”

Point #2

The results from experiment 1-3 better support and complement previous findings by Aymerich-Franch et al. and Heydrich et al. - and support the conclusion that people can feel ownership for more than one virtual body, under specific conditions. However, the most novel and interesting question is whether people can genuinely experience to be in 2 locations at the same time. This was really tested in Experiment 5, as in experiment 4, the two bodies were anyway shown close to each other, although in split halves of the screen (but still potentially perceived within one’s own PPS). Unfortunately, results from Experiment 5 are not conclusive. Questionnaire scores reported higher ratings for self-location items in the synchronous vs. asynchronous conditions, but ratings were generally low (around 1). Now, these illusions are normally mild, and it is normally to not find very higher scores, but unfortunately this single very mild effect is the only one supporting the hypothesis of a shift in self-location. Indeed, results from the self-location task are not conclusive: the task is performed at the end of a 3-minutes long stimulation alternating between the two points of view. It is possible that the participants report high scores both for location A and for location B by alternating from one to another, as a function of the stimulation, without never experiencing being in two places at the same time.

Authors’ response: We agree with the reviewer that the results of experiment 5 are inconclusive, and should be considered as hypothesis generating. We already acknowledge this in the discussion: *“It should be noted that there are several limitations of experiment 5, including the lack of objective behavioral evidence for the ownership illusion (SCR) or a control for truly simultaneous*

ownership and self-location versus switching back and forth between the two illusory bodies. The results should therefore be considered preliminary and interpreted with caution, and should serve as hypothesis generating for future studies examining the minimal conditions required for eliciting dual self-location.” In our view, further extending the results of experiment 5, with additional control experiments and SCR measurements, is beyond the scope of this study, which already contains a wealth of data from five separate experiments. That said, we believe that the results of experiment 5 are sufficiently interesting to the field, in terms of hypothesis-generating potential, to be included in the manuscript, and we therefore chose to include the results in this publication instead of them ending up in a drawer.

Point #3

Self location results (5B). There is something surprisingly in the self-location ratings. One should expect higher self location at location of the physical body, at least in the asynchronous conditions. Then eventually these ratings should increase for the two avatars' location in the synchronous condition. Instead, the ratings are relatively low for the physical body. Does this depend on just showing a different point of view from the HMD? I am wondering whether the authors collected a “baseline” rating, with no HMD, or HMD on but no body being shown, to make sense of this relatively lower self-location within the physical body.

Authors' response: We thank the reviewer for bringing this aspect of the self-location results in experiment 4 to our attention. First of all, we would like to emphasize that the results of experiment 4 were negative: we did not find support for our hypothesis that a “split 1PP” in conjunction with synchronous visuotactile stimulation would lead to dual self-location. There was no meaningful difference between the synchronous and asynchronous conditions on self-location ratings, and we therefore concluded that *“These findings suggest that this ‘split IPP’ version of the dual full-body illusion cannot be used to manipulate the sense of dual self-location in a predictable manner.”* In light of this, we do not consider it surprising that subjects experience some self-location (in both SS and AA conditions) at the position of the cameras, and some self-location at the physical location.

Point #4

The paper contains 5 experiments, whose rationale is explained in a long method section, pp. 10-14, which presents in details each experiment. When reading the results, the reader has to go back to this section to properly understand what the authors are testing in each result section. I suggest to add to each results section a brief paradigm to remind the reader the key hypothesis tested / manipulation for that specific experiment.

Authors' response:

As suggested, to improve the clarity of the manuscript, we have added reminders of the aim and paradigm of each experiment in their respective results section. To not run at the risk of being overly repetitive, we kept these reminders short and concise.

Point #5

Results, Experiment 2-3 pp. 14-15 : Add a comment / explanation to Figure 3B, to better link to Experiment 3.

Authors' response:

We have added a relevant comment to remind the reader the aim of Experiment 3 below Fig 3. The caption now reads: "Threat-evoked SCRs in experiment 2 (panel A) and experiment 3 (panel B). Experiment 2 aimed to establish the presence of body ownership illusion over two bodies, while experiment 3 served as an additional control on whether dual body ownership is truly simultaneous, or, rather, if single-body ownership transitions between the left and right illusory bodies. Error bars represent the standard error. * $p < 0.05$. ** $p < 0.01$."

Point #6

Page 17-18: "However, to the best of the authors' knowledge, there are no known cases of brain-lesion induced full-body phantoms, defined as a phantom duplication of the patient's body from the neck and down". This is actually the definition of "he-autoscopy" (see e.g., Brugger, P., Agosti, R., Regard, M., Wieser, H., and Landis, T. (1994). Heautoscopy, epilepsy and suicide. J. Neurol. Neurosurg. Psychiatry 57, 838–839.). There is no single clear brain lesion resulting in this phenomenon, but for sure it has been reported.

Authors' response:

We thank the reviewer for bringing this up to our attention. We have reviewed the literature on he-autoscopic cases and have updated the relevant paragraph on page 17, as well as our references. In particular, we commented on its very unclear etiology and its association with different neurological and psychiatric conditions, such as schizophrenia, astrocytoma in the temporal lobe, major depression episodes or epilepsy. The paragraph now reads: "Patients reporting the experience of full-body phantoms, so-called he-autoscopy, are more rare, and has been associated with strikingly different conditions (epilepsy [63], astrocytoma in the insular region of the left temporal lobe [64], schizophrenia [65], or major depression [66]). Some he-autoscopic experiences are described as a phantom duplication of the patient's body from the neck and down."

Reviewer 2

This manuscript investigates whether the experience of dual body-ownership and self-location can be induced through sensory stimulation and perceptual illusions. In a natural development from previous virtual reality studies, the experiment deploys comprehensive methods to probe the mechanisms that account for a supernumerary self. The techniques are solid and the results are easy to understand. Despite that, there are a few points I'd like to address:

Authors' response:

We are grateful for the reviewer's insightful and constructive comments, which have contributed to a substantial improvement of the manuscript. Please find our responses below.

Point #1

Is there a reason for the number of participants chosen (pp 6 line 49) based on the referenced study?

Authors' response: The reference study (Guterstam et al 2013) had a similar methodological approach as the present study, and used these sample sizes (twenty subjects for questionnaire experiments, and 30 subjects for SCR experiments) and was able to detect meaningful and differences between test and control conditions consistently across nine experiments. We therefore considered these sample size targets as suitable for the current study.

Point #2

On page 6, "Visual stimuli and head mounted displays", there is no mention of the focal length of the video the subject watched, the perceived angle of view or the display's field of view. It would also be interesting to clarify if head movement had any effect on the video (and to what degree) or if the video was fixed position-wise. Since the video contains objects moving relatively fast in a fixed background (the tactile stimuli) the frame rate and display refresh rate used during the experiment should also be mentioned. These factors influence the feeling of immersion and presence, which could affect the sense of body ownership. Although small artifacts or a low refresh rate may not hinder single body illusions, the decrease in immersion could affect more complex illusions (such as dual body); which could also account for different results in other experiments. Besides, this information is essential to replicate the study.

Authors' response:

We thank the reviewers for bringing this up, and have added additional information to the section "Visual stimuli and head mounted displays". Notably, the videos were recorded at 60

frames per seconds. The refresh rate of the head mounted displays showing the videos was 75 Hz. The videos depicting a first person perspective were recorded with the cameras positioned at the level of the body's eyes and directed towards the feet. The videos were viewed with the participant's head slightly tilted to look down towards their feet, matching the first person perspective of the videos. The perspective in the videos was fixed and unaffected by head movement, so participants were asked to keep their head still throughout the session so as to not break the immersion.

Point #3

Page 9 line 8 refers to "Fig. 1A", but figure 1 has no clear subdivisions.

Authors' response:

We thank the reviewer for bringing this to our attention and have updated Fig 1 with clear A and B subdivisions.

Point #4

Page 9 lines 15-21: both conditions described are the same, although the acronyms (SA, AS) are correct: "(3) synchronous stimulation the real body and the left illusory body but asynchronous stimulation right illusory body (SA), (4) synchronous stimulation the real body and the left illusory body but asynchronous stimulation right illusory body (AS)."

Authors' response:

We thank the reviewer for pointing out this mistake. The typo has been corrected and the section now reads "(3) synchronous stimulation the real body and the left illusory body but asynchronous stimulation right illusory body (SA), (4) synchronous stimulation the real body and the right left illusory body but asynchronous stimulation left right illusory body (AS)"

Point #5

Page 10 lines 15-28: Could attention affect the likeness of the body being owned? The synchronized stimuli may draw attention to one body over the other and attention is a process intertwined with conscious perception. When both bodies were displayed at the same time, eye recording would be able to tell if attention to each body was asymmetric (and dependent on stimulus synchronization).

Authors' response:

We thank the reviewer for this significant remark and have included a statement on potential interplays of spatial attention and dual-body illusion. Although we did not measure eye

position nor explicitly controlled for visuo-spatial attention in this study, we have done so in two previous studies investigating related body ownership illusions (Guterstam et al 2015 *Current Biology*; Gentile et al 2013 *J Neuroscience*). In Guterstam et al 2015, we did not find any meaningful differences in eye movements across synchronous illusion conditions and asynchronous control conditions. In Gentile et al 2013, the fMRI activations [synchronous vs asynchronous] in multisensory areas were unaffected by the presentation of a simultaneous visuo-spatial attention task that explicitly controlled for participants' spatial allocation of attention. In light of these previous results, we consider it unlikely that the current results would be fully explained by differences in attention or eye movements.

It is possible that dual-body illusion setups, in contrast to single-body or single-limb setups, are associated with an interaction between attention allocation and body-ownership probability. In the revised *Discussion* section, we have therefore included a paragraph elaborating on potential future directions, including the suggested eye movements recording. The added paragraph now reads: "It should be noted that there are several limitations of experiment 5, including the lack of objective behavioral evidence for the ownership illusion (SCR) or a control for truly simultaneous ownership and self-location versus switching back and forth between the two illusory bodies. The results from experiments 4 and 5 should therefore be considered preliminary and interpreted with caution, and should serve as hypothesis generating for future studies examining the minimal conditions required for eliciting dual self-location. Moreover, in all five experiments participants were merely passively experiencing an illusion; they had no ability to manipulate the virtual bodies in space, and we did not explicitly control their focus of attention. Future studies should investigate to what extent dual ownership allows for independent control of two bodies at once, and whether the illusion is constrained by limited resources of spatial attention (as measured by, for instance, eye tracking)."

Point #6

Page 12 lines 26,27: were the participants naive to their real position only on experiment 5? Was the lack of significant difference in the results between left and right bodies enough to affirm that knowing their real position had no effect in the illusion in the other experiments?

Authors' response: Yes. This information has been added to the methods section of experiment 4: "*In contrast to experiment 5 (see below), participants were not naive to the placement of the beds and cameras in the experiment room.*"

Point #7

Page 14 line 19: I assume the figure called is 3B instead of 4B.

Authors' response:

We have made the correction on page 14.

Point #8

Page 15 line 15: Statement S4, not S5.

Authors' response:

We have made the correction on page 15.

Point #9

Page 15: lines 12-19: The affirmation that "the SS condition was associated with a stronger sense of dual body ownership than AA across the three different visual perspective dynamics" is misleading. Although participants indeed rated higher dual body ownership questions in SS over AA conditions, the results show numbers close to 0 (0.1, 0.3 and -0.1 respectively, lines 17-19), which indicated "I do not know, I can neither agree nor disagree" on the scale (page 7 line 22). This could be clarified in the text. I also wonder if the participants relied on the scale's visual proportions instead of the semantics of the score.

Authors' response: We thank the reviewer for this suggestion. The relatively low absolute ratings on the SS condition has now been clarified in the text: "*(although it should be noted that the absolute mean ratings in SS were close to zero, suggesting a weak illusion effect)*".

Point #10

Page 16 lines 33-38: Confusing sentence: "Our initial working hypothesis that splitting the visual field of the IPP into two equally sized halves (left and right) and placing each of the owned illusory bodies in separate spatial environments was falsified in experiment 4." Which working hypothesis was falsified, experiment 4 (dual self location in distinct environments)?

Authors' response:

We have clarified the sentence, which now reads: "Our initial hypothesis, that dual self-location would be induced by splitting the visual field of the 1PP into two equally sized halves (left and right) and placing each of the owned illusory bodies in separate spatial environments, was falsified in experiment 4. "

Point #11

Page 20 lines 21-22: *"We speculate that different groups of place cells [62] might simultaneously be active to represent the locations of each of the two bodies at the same time."* This is an interesting hypothesis that conflicts with the more common idea that hippocampal place cells are very selective and involved in differentiating instead of integrating overlapping spatial contexts. References with experimental evidence may be necessary to improve the credibility of this speculation.

Authors' response: We agree with the reviewer that this statement is highly speculative. We have therefore included a reference demonstrating that place cells can underlie these integratory functions and that they can reliably underpin robust self-localization (Strosslin, Sheynikovich, Chabbarriaga & Gernster, 2005). We also added the following sentence: *"This contrasts with the more common idea that hippocampal place cells are very selective and involved in differentiating instead of integrating overlapping spatial contexts [67]."*

Point #12

Figure 7: *On the graph legend, I suggest avoiding mixing + and - symbols when these have meanings other than addition and subtraction. Since - is already used on the text, maybe changing + to another symbol may make the graph clearer without compromising the concordance with the main text. For example: SS(1PP-1PP) instead of SS+1PP-1PP.*

Authors' response: Thank you for this suggestion. The graph legend of figure 7 has now been changed.

Point #13

The figures lack a description of what is being shown. For example, the main hypothesis of the respective experiment, the purpose of what is being shown, or the main conclusion the figures present.

Authors' response:

We have added a brief reminder of the hypothesis and aim of each experiment below their respective figures.

Point #14

The bottom legend on figures 2, 5A, and 7 about the class of statements could have a more clear divider. For example, in figure 2, without checking the statements table, it is difficult to tell if "control statements" are only S8,S9 or S7-S10.

Authors' response: We agree with the reviewer that a clear divider would improve the clarity of these figures. Figures 2, 5A and 7 and have been updated accordingly.

Point #15

Grammar: Page 9 line 29 (only the SS condition was be associated with the). Page 11 line 10 (the participants gave a verbal rating of for each statement)

Authors' response:

We have made these corrections on page 9 and 11.

Reviewer 3

The paper reports several multisensory experiments, aiming to produce the experience of having two bodies. The authors conclude this experience can be induced.

Authors' response:

We thank the reviewer for his/her insightful and constructive comments, which have contributed to a substantial improvement of the manuscript. Please find our point-by-point responses below.

Point #1

Strengths of the study include the number of participants and number of studies. A weakness lies in the way experience is measured. Questionnaire measures of illusions may involve an element of suggestion, and can induce task demands that experimenters should ideally avoid. Threat responses are informative, but do not provide much information about the underlying mechanism. The self-location measures here seem to be responses to questionnaire items, rather than direct judgements about the agent's location within a field. Thus, when participants apparently report being in two places at once, it is unclear whether this is linked to the way the question is asked (people are often inconsistent in reply to questionnaire items where one might expect consistency, e.g., "How anxious are you?"/"How worried are you?").

Authors' response:

We thank the reviewer for this relevant remark. First, we would like to emphasize that the primary aim of this study was to provide proof-of-concept that a perceptual dual-body ownership illusion is possible. In line with an established standard approach in the field, we used questionnaires in combination with threat-evoked physiological responses. To reduce the element of suggestion, we included control statements in the questionnaires, as well as experimental control conditions. We made sure to emphasize this in the *Methods* section (in particular, S7-S10 served as controls for suggestibility). Further, our key conclusions are based on differences when contrasting the SS condition to the AA, AS and SA conditions, and such difference scores are unlikely to reflect suggestions. If the observed dual-body ownership effect was entirely the result of the formulation of the key questionnaire items and/or task demands, one would expect high ratings not only in the synchronous illusion condition, but also in the asynchronous control conditions, and for the control statements. This was not the case.

Threat-evoked physiological responses are not under voluntary control and are thus inherently less susceptible to the element of suggestion. It should be noted that threat-evoked responses here were not used to provide information about the underlying neural mechanism, but merely as an indirect (non-subjective) proxy of body ownership. If a participant truly experiences a body or limb as their own, their emotional physiological system should react greater to physical

threats directed at the owned body. The observation the subjective questionnaires and physiological responses show the same pattern of results, strengthen the conclusiveness of the results.

This work provide proof-of-concept the dual-body ownership is possible, and will hopefully serve as motivation for future studies to explore potential brain mechanisms in more detail.

Point #2

In expts 1-3, the participant views two virtual sticks stroking two different virtual legs. The viewpoint seems to be shifted towards the middle of the two visual bodies. One virtual body has their left leg stimulated, and the other virtual body as their right leg stimulated. However, it is unclear whether tactile stimulation is applied to both legs - this should be in the Methods.

Authors' response:

We have clarified the exact way the stimuli were applied to participants' body parts in the *Visual stimuli and head-mounted displays* section. As shown in Fig 1A, when synchronous stimulation was applied, the same body part (e.g. left leg) of both the left and right bodies were touched simultaneously. There was never a situation when different limbs on the two bodies, e.g. the left leg on one virtual body and the right leg on the other virtual body, were stimulated at the same time. We have now included an additional statement clarifying that only one limb, of the same laterality in both virtual bodies, were stimulated at a time.